# School-age outcomes among IVF-conceived children: A population-wide cohort study

**Amber L. Kennedy**[1,2], **Beverley J. Vollenhoven**[3,4,5], **Richard J. Hiscock**[1,2], **Catharyn J. Stern**[1,6,7], **Susan P. Walker**[1,2], **Jeanie L. Y. Cheong**[1,8,9], **Jon L. Quach**[1,8], **Roxanne Hastie**[1,2], **David Wilkinson**[2,10], **John McBain**[1,6,7], **Lyle C. Gurrin**[11], **Vivien MacLachlan**[5], **Franca Agresta**[7], **Susan P. Baohm**[5,10], **Stephen Tong**[1,2‡], **Anthea C. Lindquist**[1‡*]

**1** Department of Obstetrics and Gynaecology, University of Melbourne, Melbourne, Australia, **2** Mercy Perinatal, Mercy Hospital for Women, Heidelberg, Victoria, Australia, **3** Department of Obstetrics and Gynaecology, Monash University, Clayton, Victoria, Australia, **4** Department of Obstetrics and Gynaecology, Monash Health, Clayton, Victoria, Australia, **5** Monash IVF, Clayton, Victoria, Australia, **6** Reproductive Services Unit, The Royal Women's Hospital, Parkville, Victoria, Australia, **7** Melbourne IVF, East Melbourne, Victoria, Australia, **8** Murdoch Children's Research Institute, Parkville, Victoria, Australia, **9** Department of Neonatology, The Royal Women's Hospital, Parkville, Victoria, Australia, **10** City Fertility Centre, Melbourne, Victoria, Australia, **11** Melbourne School of Population and Global Health, University of Melbourne, Melbourne, Australia

‡ These authors are joint senior authors on this work.
\* anthea.lindquist@unimelb.edu.au

**Data Availability Statement:** Data for this study was provided by various data custodians and linked by the Centre for Victorian Data linkage (https://www.health.vic.gov.au/reporting-planning-data/the-centre-for-victorian-data-linkage). With relevant ethical approval, data are available upon request to the governing data custodians.

## Abstract

### Background

In vitro fertilisation (IVF) is a common mode of conception. Understanding the long-term implications for these children is important. The aim of this study was to determine the causal effect of IVF conception on primary school-age childhood developmental and educational outcomes, compared with outcomes following spontaneous conception.

### Methods and findings

Causal inference methods were used to analyse observational data in a way that emulates a target randomised clinical trial. The study cohort comprised statewide linked maternal and childhood administrative data. Participants included singleton infants conceived spontaneously or via IVF, born in Victoria, Australia between 2005 and 2014 and who had school-age developmental and educational outcomes assessed. The exposure examined was conception via IVF, with spontaneous conception the control condition. Two outcome measures were assessed. The first, childhood developmental vulnerability at school entry (age 4 to 6), was assessed using the Australian Early Developmental Census (AEDC) ($n$ = 173,200) and defined as scoring <10th percentile in ≥2/5 developmental domains (physical health and wellbeing, social competence, emotional maturity, language and cognitive skills, communication skills, and general knowledge). The second, educational outcome at age 7 to 9, was assessed using National Assessment Program–Literacy and Numeracy (NAPLAN) data ($n$ = 342,311) and defined by overall z-score across 5 domains (grammar and punctuation,

**Funding:** This work was supported by the National Health and Medical Research Council through the Australian Federal Government Graduate Research Scheme (AK) and Mercy Foundation, through Mercy Perinatal (AK). Ferring Pharmaceutics supported this work through an unconditional research grant (AK). The funders had no role in the study design, data collection and analysis, decision to publish, or preparation of the manuscript.

**Competing interests:** I have read the journal's policy and the authors of this manuscript have the following competing interests: BV has a paid role as a member of the Therapeutic Goods Administration. BV, FA and KS own shares in respective IVF companies (Monash IVF, Virtus Health and Melbourne IVF).

**Abbreviations:** AEDC, Australian Early Development Census; ATE, average treatment effect; ATSI, Aboriginal and Torres Strait Islander; BMI, body mass index; CI, confidence interval; CVDL, Centre for Victorian Data Linkage; DAG, directed acyclic graph; ICSI, intracytoplasmic sperm injection; IPW, inverse probability weight; IPWRA, inverse-probability-weighted regression adjustment; IVF, in vitro fertilisation; MD, mean differences; NAPLAN, National Assessment Program–Literacy and Numeracy; NMS, national minimum standard; POM, potential outcome means; PS, propensity score; RD, risk difference; RR, relative risk; SAP, statistical analysis plan; SE, standard error; SEIFA, Socio-Economic Indexes for Areas; TMLE, targeted maximum likelihood estimation; VPDC, Victorian Perinatal Data Collection.

reading, writing, spelling, and numeracy). Inverse probability weighting with regression adjustment was used to estimate population average causal effects.

The study included 412,713 children across the 2 outcome cohorts. Linked records were available for 4,697 IVF-conceived cases and 168,503 controls for AEDC, and 8,976 cases and 333,335 controls for NAPLAN. There was no causal effect of IVF-conception on the risk of developmental vulnerability at school-entry compared with spontaneously conceived children (AEDC metrics), with an adjusted risk difference of −0.3% (95% CI −3.7% to 3.1%) and an adjusted risk ratio of 0.97 (95% CI 0.77 to 1.25). At age 7 to 9 years, there was no causal effect of IVF-conception on the NAPLAN overall z-score, with an adjusted mean difference of 0.030 (95% CI −0.018 to 0.077) between IVF- and spontaneously conceived children. The models were adjusted for sex at birth, age at assessment, language background other than English, socioeconomic status, maternal age, parity, and education. Study limitations included the use of observational data, the potential for unmeasured confounding, the presence of missing data, and the necessary restriction of the cohort to children attending school.

## Conclusions

In this analysis, under the given causal assumptions, the school-age developmental and educational outcomes for children conceived by IVF are equivalent to those of spontaneously conceived children. These findings provide important reassurance for current and prospective parents and for clinicians.

## Author summary

### Why was this study done?

- More than 8 million children have been conceived globally with the assistance of in vitro fertilisation (IVF).

- Some studies suggest these children have an increased risk of congenital abnormalities, autism spectrum disorder, developmental delay, and intellectual disability.

- Educational and school-age developmental outcomes following IVF conception have not yet been adequately characterised.

### What did the researchers do and find?

- Using statewide, linked population data from Victoria, Australia, we investigated the school-age developmental and educational outcomes for children born following IVF-assisted conception.

- The study examined 2 separate assessments of school-age development and educational outcomes among 585,659 children, including 11,059 children who were conceived via IVF.

- This study was designed and performed within a causal framework, in order to produce the best possible estimate of exposure effect using observational data.

- We found no difference in school-age childhood developmental and educational outcomes between IVF- and spontaneously conceived children.

### What do these findings mean?

- These findings provide reassurance for current and prospective parents, as well as clinicians who are involved in IVF.

- This information may be useful in providing informed consent and education to those considering IVF and those with children conceived via IVF.

## Introduction

In vitro fertilisation (IVF) is a common mode of conception worldwide [1]. Since the first successful IVF birth in 1978, more than 8 million babies have been born globally following IVF conception [2,3]. In Australia, it is now estimated that 1 in 20 babies are born following IVF conception [4,5].

As the number of children born following IVF conception continues to rise, a deeper understanding of the long-term implications for these children is important. It is well established that there are increased risks of maternal and perinatal complications following IVF conception [6–8]. Large cohort studies have suggested an increase in the frequency of congenital abnormalities, autism spectrum disorder, developmental delay, and intellectual disability in children conceived via IVF or intracytoplasmic sperm injection (ICSI) techniques [9–13]. However, reports detailing longer term outcomes after IVF beyond the neonatal period remain sparse.

Educational and cognitive outcomes following IVF conception have not yet been thoroughly investigated. Several small cohort studies [14–17] have reported conflicting results. One large population-based study suggested a small difference in school performance in favour of spontaneous conception [18]. Another population study recently concluded that school performance was not adversely affected by the process of IVF but, rather, the condition of subfertility [19].

Parents of IVF- and spontaneously conceived children possess inherently different health and sociodemographic characteristics [20,21]. Factors such as increased maternal age and higher education are known to be associated with both the use of fertility treatment and better early childhood outcomes [22–24]. It is thus critical that such factors are appropriately acknowledged when examining the association between mode of conception and childhood outcomes. Proper adjustment in any statistical analysis is required before any association can be given a causal interpretation.

Our study aimed to overcome some of the limitations of the analysis of observational (non-randomised) data by using a causal inference approach that seeks to emulate the results of a randomised comparison in a clinical trial (Table 1) [25,26]. This analytical approach attempts to simulate a randomised trial by (1) requiring an a priori statistical analysis protocol; (2)

addressing a causal question reflecting the effect of an intervention at a specific clinical decision point on a prespecified outcome; and (3) using inverse probability weighting via propensity score (PS) models to balance the outcome propensity differences between exposed and control populations, with the aim of producing exchangeable comparison groups [27]. This allowed us to estimate the population-average effect of mode of conception (IVF versus spontaneous conception) on childhood developmental and educational outcomes with a causal interpretation. Our study aims to estimate the total causal effect of IVF conception on school-age childhood developmental and educational outcomes using a causal inference approach and employing the necessary assumptions.

## Methods

### Study design

This study is reported as per the Strengthening the Reporting of Observational Studies in Epidemiology (STROBE) guideline (Checklist in S13 File).

**Population.** The study population included all singleton livebirths in Victoria between 2005 and 2014. Twins and higher order multiple births were excluded. Perinatal information

**Table 1. Target trial emulation.**

| Protocol component | Research question: What is the effect of mode of conception (IVF/non-IVF) on childhood development? | |
|---|---|---|
| | **Target trial** | **Emulation** |
| **(i) Eligibility criteria** | Inclusion criteria: All couples wanting to conceive and with the ability to conceive by either method. Exclusion criteria: Couples either without the ability to conceive or can only conceive using IVF. E.g., maternal age >45 not compatible with spontaneous conception. | Inclusion criteria: Same as target trial. **Limitation: It is not possible to collect data on failed attempts, miscarriages, or stillbirths = Livebirth bias. Livebirth bias introduces potential selection bias, collider bias, and "depletion of susceptible cases" (see below).** Exclusion criteria: Same as target trial, same exclusion criteria can be applied—important to ensure positivity is maintained between exposure groups. |
| **(ii) Treatment strategies** | (A) Spontaneous ("in vivo") conception leading to a live birth (B) Conception aided by IVF leading to a live birth | (A) Same as target trial (B) Same as target trial **Positivity assumption: Every subject could potentially be included in either exposure group.** This is both a design feature as well as one of statistical adjustment where the positivity assumption is addressed by dataset trimming to ensure overlapping of inverse probability weights. |
| **(iii) Assignment procedures** | Randomised at entry—decision to conceive vs. Modified intention to treat (ITT)—randomised at decision to conceive and included in trial after subsequent live birth. | Modified ITT—time commences from livebirth. DAG used to identify prespecified covariates to be included in estimator models for both confounding control and outcome adjustment. Estimator used a doubly robust inverse-probability-weighted regression adjustment model to achieve adherence to ignorability and positivity assumptions, such that observed groups can be considered balanced or exchangeable. |
| **(iv) Follow-up period** | AEDC–school entry, 4–6 years of age NAPLAN–Grade 3, 7–9 years of age Some trial participants would be lost to follow-up. | Same as target trial. **Considerations and limitations**: Post-exposure and pre-selection: live birth bias Post-exposure and post-selection: follow-up divided into (1) postnatal loss (akin to loss to follow-up), and (2) unobserved confounding. (1) Postnatal: This is distinct from livebirth bias inclusion loss to follow-up post "live birth" due to neonatal death, infant death, childhood death, non-school attendance, missing outcome data. (i) Dataset does not contain child death data (iia) Missing data due to disability to be analysed as sensitivity analyses (AEDC: special needs, NAPLAN: exempt) (see below under outcome) (iib) Dataset does not contain data on children who do not attend school, i.e., severe disability—limitation for discussion. (2) Unobserved confounding: mediators (post-exposure) vs. unmeasured (pre-exposure) confounders. Sensitivity analysis for unmeasured confounders: E values were calculated. |

*(Continued)*

**Table 1.** (Continued)

| Protocol component | Research question: What is the effect of mode of conception (IVF/non-IVF) on childhood development? | |
| --- | --- | --- |
| | **Target trial** | **Emulation** |
| **(v) Outcome:** | **AEDC–Binary**<br>• Primary outcome: DV2 –developmental vulnerability in ≥2 domains<br>• Secondary outcome: individual domains<br>**NAPLAN—Continuous**<br>• Primary outcome: overall score (z-score)<br>• Secondary outcome: individual domains<br>**NAPLAN—Binary**<br>• Below national minimum standard in individual domains | Same outcome measures as target trial.<br>**Considerations and sources of bias:**<br>**AEDC**<br>• Children commence school at different ages at the start of calendar year and the assessment is not standardised for age range (4–6); age of assessment will be included in model for standardisation.<br>• Year of assessment is not included in model (i.e., 2012, 2015, 2018), as the metric is the same every year.<br>Children defined as having special needs are not awarded a valid result for the AEDC domains, however, by definition, are developmentally vulnerable, thus "missing not at random." Methods for addressing "special needs status":<br>(1) Primary analysis—"special needs missing" coded as "vulnerable." Therefore, no missing outcome data or exposure data and imputation of missing covariate data only performed.<br>(2) Sensitivity analyses: (a) excluding special needs cases completely and (b) imputing their outcomes (biased); this data is MNAR and thus this is performed for illustration.<br>**NAPLAN**<br>Same NAPLAN outcome measure as target trial.<br>• Children commence school at difference ages at the start of calendar year and the assessment is not adjusted for age range (7–9), thus age of assessment will be included in model for standardisation.<br>• The NAPLAN paper is different each year, thus year of assessment is included in model.<br>Children exempt from sitting NAPLAN are, by definition, below the NAPLAN national minimum standard for each domain from which they have been excluded, thus "missing not at random." Methods for addressing NAPLAN "exempt" status:<br>(1) Primary analysis—exempt coded as either the lowest possible test z-score (continuous) or below the national minimum standard (binary) NAPLAN domain outcomes. Therefore, no missing outcome data or exposure data and imputation of missing covariate data performed.<br>(2) Sensitivity analyses: (a) excluding exempt cases completely and (b) imputing their outcomes (biased); these data are MNAR and thus this was performed for illustration only. |
| **(vi) Causal contrasts of interest and analysis plan** | ITT or modified ITT:<br>**AEDC:**<br>RD of being developmentally vulnerable (point estimate RD 95% CI)<br>RR of being developmentally vulnerable (point estimate RR 95% CI)<br>**NAPLAN (continuous):**<br>Difference in mean (MD) standardised NAPLAN total score (point estimate MD and 95% CI)<br>**NAPLAN (binary)**<br>RR of being below national minimum standard (point estimate RR 95% CI) | **Causal comparisons:**<br>(1) Is the risk of being developmentally vulnerable different in children born to mothers who conceived via IVF compared with mothers who conceived spontaneously?<br>(2) Is the mean overall NAPLAN score different for children born to mothers who conceived via IVF compared with mothers who conceived spontaneously?<br>**Estimand = ATE**<br>**AEDC:**<br>RD of being developmentally vulnerability (ATE RD 95% CI)<br>RR of being developmentally vulnerability (ATE RR 95% CI)<br>**NAPLAN (continuous):**<br>Difference in mean standardised NAPLAN total score; (ATE MD and 95% CI)<br>**NAPLAN (binary):**<br>RR of being below national minimum standard (ATE RR 95% CI)<br>*Estimator model*—Doubly robust IPW (with regression adjustment) with robust SE for maternal clustering—some mothers represented by several children (as detailed in Methods). |

AEDC, Australian Early Development Census; ATE, average treatment effect; DAG, directed acyclic graph; IPW, inverse probability weight; IVF, in vitro fertilisation; NAPLAN, National Assessment Program for Literacy and Numeracy; RD, risk difference; RR, relative risk; SE, standard error.

was collected from audited birth outcome data through the Victorian Perinatal Data Collection (VPDC) [28,29]. The 3 largest IVF units in Victoria provided maternal records from all cycles that resulted in a birth during the study period. Creation of linked maternal/child data pairs required matching of the VPDC data with birth records, which were obtained from the Victorian Births, Deaths and Marriage registry.

**Exposure.**    The exposure was conception via IVF compared with spontaneous conception. The term "IVF" is used collectively to include both conventional IVF, IVF with ICSI, and associated laboratory techniques. IVF cases were identified through the IVF database. Victorian births not identified in the IVF database were allocated to the control group. Pregnancies recorded as "IVF conception" in the VPDC but not identified within the IVF database were excluded, ensuring the control group did not contain any IVF conceptions. These cases likely represent overseas or interstate IVF conceptions, Victorian IVF conceptions not captured by our database, or failed linkages between the IVF database and state birth records.

**Main outcome measures.**    Childhood educational and developmental outcomes were assessed using 2 standardised, national assessments. The Australian Early Development Census (AEDC) [30] and The National Assessment Program–Literacy and Numeracy (NAPLAN) [31]. See Supporting information file (Methods in S1 File), for a detailed description of each measure.

**Australian Early Developmental Census (AEDC).**    The AEDC assesses broad childhood functional development at school entry (age 4 to 6) across 5 domains: physical health and well-being, social competence, emotional maturity, language and cognitive skills (school-based) and communication skills, and general knowledge. The primary AEDC outcome for this study was a global measure, developmental vulnerability, defined as scoring $<$10th percentile in $\geq 2$ of the 5 developmental domains. The secondary outcomes included developmental vulnerability in each of the 5 domains.

**The National Assessment Program–Literacy and Numeracy (NAPLAN).**    NAPLAN is a school-based psychometric assessment, assessing 5 educational domains: grammar and punctuation, reading, writing, spelling, and numeracy [32]. The study cohort's grade 3 NAPLAN (fourth year of primary school) results were investigated. For this study, an overall z-score was calculated and used as the primary outcome, with the individual domain z-scores examined as secondary outcomes. By a priori consensus, a mean z-score difference of 0.2 standard deviations was considered to be clinically relevant. Individual domain scores below the published national minimum standard (NMS) NAPLAN scores for each year and for each domain were analysed as secondary (binary) outcomes.

**Covariates.**    Covariates to be considered for inclusion in the statistical analysis models were decided a priori by the authorship team whose expertise included epidemiology, perinatology, reproductive endocrinology, and education. These covariates included child's sex (as assigned at birth), child's age in years at assessment, language background other than English (LBOTE), maternal age (at birth of the child), parity and both maternal and paternal highest obtained level of education, and socioeconomic status [33]. Gestational age at birth, mode of delivery, and birthweight were considered to be mediators on the causal pathways of interest and therefore not adjusted for in this analysis. A directed acyclic graph (DAG) was created to describe the structure of the relationships between all variables and identify the adjustment variable set, in line with the methodology recommendations of Tennant and colleagues [34]. Our prespecified statistical analysis plan (SAP) and the DAG were agreed upon and signed off by all authors in May 2020, prior to the commencement of data analysis (Protocol in S2 File).

**Linkage.**    Administrative record linkage techniques were employed to match cases with the exposure (conception via IVF) through to childhood outcome data. Data linkage was performed by the Centre for Victorian Data Linkage (CVDL), a third-party government-funded

data linkage unit [35]. Probabilistic linkage was performed between the 5 individual databases —birth records, birthing outcomes, IVF records, AEDC, and NAPLAN. Post-linkage data were manually screened for false matches using secondary variables (e.g., residential postcode). False matches and duplicates were removed (Table A in S3 File outlines number and percentage of successful linkages).

Two separate, linked study populations were identified, children with a linked AEDC record and children with a linked NAPLAN record. These 2 cohorts were analysed and are reported separately. Some children were included in both cohorts.

## Causal assumptions

The ATE (average treatment effect) estimand used in this study is based upon the Potential Outcomes Framework. If a set of assumptions is met, then causal interpretation can be made. The causal assumptions are counterfactual consistency, ignorability (conditional exchangeability), and positivity. Counterfactual consistency means that the definition of exposure is consistent for all individuals. Ignorability states that treatment assignment can be considered random after controlling for, conditioning on, a set of covariates [36]. By identifying confounding variables, and importantly, the structure of the relationships between variables, via a DAG (S1 Fig) and by performing appropriate statistical modelling to balance the population (for example, inverse probability weighting), observed populations can be considered exchangeable or "unconfounded." Exchangeability requires that there are no important unmeasured confounders; this assertion is untestable. The positivity assumption means that for all observations, the conditional probability of being exposed (receiving treatment/no treatment) is greater than zero. This is likely violated if overlap of the control and exposed populations is poor [26].

To best emulate a target trial, it must be possible for all participants to potentially receive both treatments. To ensure the assumptions underlying our causal approach were as robust as possible, we considered our observational data in direct comparison with the conditions of a target trial (Table 1)

## Handling of missing data

The proportions of missing data are described in Table 2. Data were missing for outcomes and covariates; there were no missing exposure data. Missing covariates and outcomes that were considered to either be missing completely at random or missing at random were imputed. Children identified as having special needs are not allocated an AEDC domain category and thus their outcome data is missing. Likewise, children who attend school but have a disability that precludes them from being able to appropriately participate in the NAPLAN are exempt from sitting the test; by definition, these children are below the NAPLAN national minimum standard for each domain from which they have been excluded. Outcome data for these children for both AEDC domain categories and NAPLAN z-scores was considered missing not at random. In the analysis of all primary and secondary AEDC outcomes, children with special needs have been included and assumed to be "developmentally vulnerable." In the analysis of NAPLAN outcomes, "exempt" children have also been included and allocated either the lowest possible test z-score or deemed to be below the national minimum standard for binary NAPLAN domain outcomes.

All covariates in the analysis model that had missing data were imputed, even if missing was very low. For the AEDC analysis, imputed covariates included parity, age at assessment, maternal education, Socio-Economic Indexes for Areas (SEIFA), and outcome score. For the NAPLAN analysis, imputed covariates included parity, age at assessment, maternal education,

**Table 2. Baseline characteristics of study cohort (including cases with missing outcome and covariate data).**

| Baseline characteristic | AEDC | | | NAPLAN | | |
|---|---|---|---|---|---|---|
| | TOTAL cohort N = 173,200 | Controls N = 168,503 | IVF conceptions N = 4,697 | TOTAL cohort N = 342,311 | Controls N = 333,335 | IVF conceptions N = 8,976 |
| **Child baseline data** | | | | | | |
| Sex (% female) | 48.7 | 48.7 | 50.8 | 49.0 | 49.0 | 49.9 |
| *Missing (%)* | *0.0* | *0.0* | *0.0* | *0.0* | *0.0* | *0.0* |
| Age of assessment in years (decimal) (Median [IQR]) | 5.4 [5.2, 5.7] | 5.4 [5.2, 5.7] | 5.4 [5.2, 5.7] | 8.3 [8.1, 8.6] | 8.3 [8.3, 8.6] | 8.4 [8.1, 8.6] |
| *Missing (%)* | *<0.1* | *<0.1* | *<0.1* | *0.0* | *0.0* | *0.0* |
| Language background other than English (%) | 18.6 | 18.8 | 13.5 | 23.5 | 23.6 | 20.1 |
| *Missing (%)* | *0.0* | *0.0* | *0.0* | *0.0* | *0.0* | *0.0* |
| ATSI (%) | 2.3 | 2.3 | 0.6 | 1.0 | 1.0 | 0.2 |
| *Missing (%)* | *0.0* | *0.0* | *0.0* | *0.0* | *0.0* | *0.0* |
| Mother born overseas (%) | 27.3 | 27.4 | 23.7 | 25.1 | 25.1 | 21.3 |
| *Missing (%)* | *0.0* | *0.0* | *0.0* | *0.0* | *0.0* | *0.0* |
| Birthweight (grams) (mean (SD)) | 3,416 (544) | 3,417 (543) | 3,359 (571) | 3,422.7 (570) | 3,424.6 (569) | 3,352 (618) |
| *Missing (%)* | *<0.1* | *<0.1* | *<0.1* | *<0.1* | *<0.1* | *<0.1* |
| Small for gestational age (<10th centile)[a] (%) | 9.3 | 9.3 | 8.60 | 9.3 | 9.3 | 8.60 |
| Gestational age of delivery in weeks—decimal (Median and [IQR]) *Missing (%)* | 39.6 [38.6, 40.6] *<0.1* | 39.7 [38.7, 40.6] *<0.1* | 39.1 [38.3, 40.3] *0.2* | 39.5 [38.7, 40.7] *<0.1* | 39.5 [38.7, 40.7] *<0.1* | 39.0 [38.3, 40.3] *<0.1* |
| Mode of birth (% of each category) | | | | | | |
| • Unassisted vaginal birth | 54.9 | 55.5 | 32.7 | 55.7 | 56.3 | 33.9 |
| • Instrumental vaginal birth | 14.0 | 13.8 | 19.6 | 13.7 | 13.6 | 19.1 |
| • Cesarean section | 31.1 | 30.7 | 47.7 | 30.6 | 30.1 | 46.9 |
| *Missing (%)* | *<0.1* | *<0.1* | *<0.1* | *<0.1* | *<0.1* | *<0.1* |
| Congenital anomaly (%) | 3.7 | 3.7 | 3.8 | 3.2 | 3.2 | 3.4 |
| *Missing (%)* | *0.1* | *0.1* | *0.1* | *<0.1* | *<0.1* | *<0.1* |
| Special needs status (AEDC)/exempt from NAPLAN (%) | 5.2 | 5.2 | 5.2 | 2.1 | 2.1 | 1.4 |
| **Maternal baseline data** | | | | | | |
| Age (Median and [IQR]) *Missing (%)* | 31.4 [27.6, 35.0] *0.0* | 31.3 [27.5, 34.8] *0.0* | 35.7 [32.8, 38.6] *0.0* | 31.6 [27.7, 35.1] *0.0* | 31.5 [27.7, 35.1] *0.0* | 35.8 [32.9, 38.8] *0.0* |
| Mother born overseas (%) | 27.3 | 27.4 | 23.7 | 25.1 | 25.1 | 21.3 |
| *Missing (%)* | *0.0* | *0.0* | *0.0* | *0.0* | *0.0* | *0.0* |
| High school level of education (% for each): | 2.8 | 2.9 | 0.7 | 4.2 | 4.2 | 1.6 |
| • Year 9 or below | 6.4 | 6.5 | 3.0 | 9.3 | 9.4 | 5.1 |
| • Year 10Year | 6.4 | 6.5 | 4.3 | 10.6 | 10.7 | 8.9 |
| • 11Year 12 and above | 53.8 | 53.6 | 62.6 | 74.1 | 73.8 | 81.7 |
| *Missing (%)* | *30.5* | *30.6* | *29.3* | *1.9* | *1.9* | *2.8* |
| Post-school level of education (% for each): | 12.9 | 13.0 | 8.2 | 21.4 | 21.6 | 15.2 |
| • No post-school education | 16.9 | 17.1 | 11.1 | 23.4 | 23.6 | 15.6 |
| • Certificate (including trade) | 10.9 | 10.9 | 10.9 | 15.6 | 15.6 | 15.9 |
| • Advanced diploma | 27.7 | 27.4 | 39.9 | 36.5 | 36.2 | 49.3 |
| • Bachelor degree or above | *31.6* | *31.6* | *30.5* | *3.1* | *3.1* | *4.0* |
| *Missing (%)* | | | | | | |
| Parity (% for each category): | 43.2 | 42.6 | 64.3 | 42.8 | 42.2 | 63.7 |
| 0 | 35.1 | 35.2 | 29.6 | 34.9 | 35.1 | 28.0 |
| 1 | 14.5 | 14.8 | 4.8 | 14.9 | 15.2 | 4.8 |
| 2 | 4.6 | 4.7 | 0.9 | 4.7 | 4.8 | 1.1 |
| 3 | 1.5 | 1.5 | 0.2 | 1.5 | 1.6 | 0.2 |
| 4 | 1.1 | 1.2 | 0.1 | 1.1 | 1.1 | 0.2 |
| 5+ | *0.02* | *0.02* | *0.00* | *0.06* | *0.01* | *2.04* |
| *Missing (%)* | | | | | | |

*(Continued)*

**Table 2.** (Continued)

| Baseline characteristic | AEDC | | | NAPLAN | | |
|---|---|---|---|---|---|---|
| | TOTAL cohort $N = 173,200$ | Controls $N = 168,503$ | IVF conceptions $N = 4,697$ | TOTAL cohort $N = 342,311$ | Controls $N = 333,335$ | IVF conceptions $N = 8,976$ |
| SEIFA quintile: | 16.4 | 16.3 | 6.0 | 17.1 | 17.4 | 7.0 |
| 1 (most disadvantaged) | 18.0 | 18.2 | 11.2 | 14.2 | 14.3 | 9.8 |
| 2 | 21.94 | 22.0 | 20.2 | 20.2 | 20.4 | 14.2 |
| 3 | 23.4 | 23.2 | 28.3 | 23.5 | 23.4 | 26.4 |
| 4 | 20.2 | 19.9 | 34.2 | 25.0 | 24.5 | 42.6 |
| 5 (least disadvantaged) | | | | | | |
| *Missing (%)* | *0.1* | *0.1* | *0.1* | *0.1* | *0.1* | *0.1* |
| Occupation (% for each category): | *Not in database* | *Not in database* | *Not in database* | 17.3 | 17.1 | 26.9 |
| • Senior management | | | | 17.3 | 17.2 | 22.1 |
| • Other business manager | | | | 18.0 | 18.1 | 17.7 |
| • Tradesman/woman, clerks, sales, and service staff | | | | 12.5 | 12.7 | 7.8 |
| • Machine operators, hospitality staff, assistants, labourers, and related workers | | | | 32.6 | 32.9 | 22.8 |
| • Not in paid work | | | | | | |
| • *Missing (%)* | | | | *2.2* | *2.2* | *2.7* |
| **Second parent baseline data** | | | | | | |
| Level of education (% for each category): | 3.1 | 3.2 | 1.4 | 5.0 | 5.1 | 2.9 |
| High school: - Year 9 or below | 7.5 | 7.6 | 4.9 | 11.8 | 11.9 | 8.4 |
| • Year 10 | 6.3 | 6.3 | 5.8 | 11.6 | 11.6 | 11.1 |
| yYear 11 | 40.9 | 40.6 | 51.3 | 57.8 | 57.6 | 67.2 |
| • Year 12 and above | | | | | | |
| *Missing (%)* | *42.2* | *42.3* | *36.7* | *13.8* | *13.9* | *10.4* |
| Level of education (% for each category): | *Not in database* | *Not in database* | *Not in database* | 18.1 | 18.2 | 14.4 |
| • Post-school: - No post-school education | | | | 27.8 | 27.9 | 22.2 |
| • Certificate (including trade) | | | | 11.1 | 11.1 | 12.7 |
| • Advanced diploma | | | | 27.7 | 27.4 | 38.8 |
| • Bachelor degree or above | | | | | | |
| *Missing (%)* | | | | *15.4* | *15.5* | *11.9* |
| Occupation (% for each category): | *Not in database* | *Not in database* | *Not in database* | 18.5 | 18.2 | 29.7 |
| • Senior management | | | | 21.7 | 21.6 | 26.7 |
| • Other business manager | | | | 22.2 | 22.3 | 19.4 |
| • Tradesman/woman, clerks, sales, and service staff | | | | 16.6 | 16.7 | 9.9 |
| • Machine operators, hospitality staff, assistants, labourers, and related workers | | | | 8.1 | 8.1 | 5.0 |
| • Not in paid work | | | | | | |
| *Missing (%)* | | | | *12.9* | *13.0* | *9.3* |

AEDC, Australian Early Development Census; ATSI, Aboriginal and Torres Strait Islander; IVF, in vitro fertilisation (cases); IQR, interquartile range; NAPLAN, National Assessment Program for Literacy and Numeracy; SD, standard deviation; SEIFA, Socio-Economic Indexes for Areas by residential postcode.

[a] Birth weight centile derived from Dobbin's growth chart.

paternal education, SEIFA, and outcome score. Maternal body mass index (BMI) was excluded from imputation and analysis because the missingness was too high (>50%). For AEDC imputation models, second parent education level (42% missing) was excluded due to non-convergence when included in the imputation model.

Multiple imputation of missing data was performed under a fully conditional specification using a predictive mean model for continuous and unordered categorical covariates and a logistic model for binary covariates, with standard errors (SEs) accounting for maternal clustering (Methods, Table 2 and Figs A–C in S5 File). The model contained outcome, exposure, model covariates, and auxiliary variables (AEDC: remote locality, Aboriginal and Torres Strait Islander (ATSI) status, and maternal country of origin; NAPLAN: ATSI status and maternal

country of origin) along with interaction terms (exposure-parity, exposure-maternal age, exposure-age at assessment, gender-age at testing) and 1 higher order term (test age$^2$). At 20 imputations, the Monte Carlo errors were less than 10% of corresponding SE for all covariates. Each imputation model was subjected to the recommended diagnostic tests [37].

## Statistical analysis

Descriptive statistics were calculated and are reported for each cohort by IVF exposure status, according to type and distribution of data.

**Treatment effect size modelling.** All multivariate models were adjusted for the listed covariates identified in the prespecified SAP, except for (1) maternal BMI; and (2) second parent education level, for AEDC outcome models only.

For each of the imputed datasets, the predicted probability of exposure or PS and associated inverse probability weight (IPW = 1/PS) were estimated using a logistic regression model, conditional on all analysis model covariates [38]. These weights were then stabilised by including as a factor in the numerator the proportion of each treatment group within the population, i.e., the prevalence of IVF and spontaneous conception [39]. Diagnostic tests performed after planned treatment effect modelling (Figs A–D in S6 File) demonstrated poor overlap of exposed and non-exposed cohorts. We therefore restricted our analysis to the IVF population whose weights overlapped with the control group to ensure that the overlap ("positivity") assumption was not violated. This reduced the IVF cases to 31.6% of the original AEDC cohort and 22.3% of the NAPLAN cohort. For each covariate, the standardised mean difference between the exposure arms was calculated to assess if balance between weighted pseudo-populations was achieved (Figs A and B in S8 File).

For each imputed dataset, a doubly robust inverse-probability-weighted regression adjustment (IPWRA) model [38,39] was then used to estimate the respective potential outcome means (POM) followed by (1) the risk difference (RD) and relative risk (RR) for binary outcomes (AEDC and NAPLAN); and (2) mean differences (MD) for continuous outcomes (NAPLAN z-score).

Finally, estimates for each imputed dataset were pooled to provide overall ATE with associated 95% confidence limits using Rubin's method.

Provided the assumptions outlined above are satisfied, the estimates generated from these analyses can be interpreted as the population average causal effect, that is, the mean effect on the outcome if the treatment was applied to the entire population and contrasted with the outcome if the entire population received the control condition.

**Clustering.** Clustering of data within mothers due to more than 1 singleton birth during the study period was accounted for in the imputation models, the calculation of inverse probability weights and estimation of the treatment effect by using robust SEs.

**Sensitivity analyses.** Sensitivity analyses were also performed to address identified sources of potential bias. For both AEDC (special needs status) and NAPLAN (exempt status) cohorts, sensitivity analyses were performed: (1) by excluding these cases completely; and (2) by imputing their outcomes (Fig A in S4 File). Targeted maximum likelihood estimation (TMLE) modelling (a machine learning ensemble that is less sensitive to violations of positivity and does not require data distribution assumptions) was undertaken for comparison [40]. Additionally, calculation of E-values for our 2 primary outcomes was performed to quantify the magnitude of unobserved bias required to alter our findings.

The analysis for this study was performed using STATA MP Version 17.0 [41], EMTLE package [40], and R [42].

### Ethics/Governance

Ethical approval for the project was obtained from Mercy, Monash Health and Melbourne IVF Health Human Research Ethics Committees. Each data custodian provided contractual approval for data access and data linkage. The CVDL approved the project and performed the linkage.

## Results

The total cohort included 585,659 singleton births in Victoria between 2005 and 2014. Among this cohort, 173,200 children, including 4,697 IVF births, were linked to AEDC outcome data. Additionally, 342,331 children, including 8,976 IVF births, were linked to NAPLAN data (Fig 1). Overall, a total of 11,059 IVF-conceived children and 401,654 spontaneously conceived children

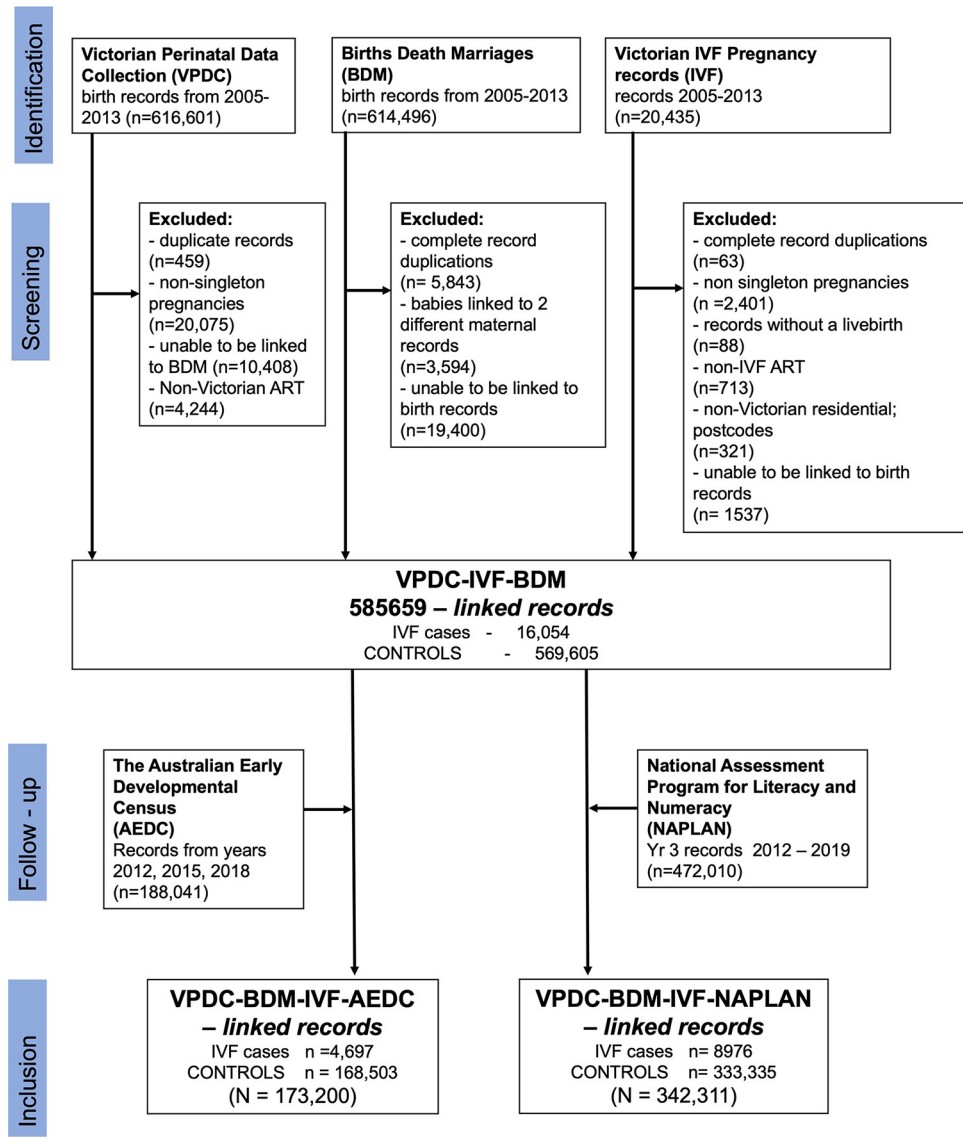

**Fig 1. Participant flow chart.** IVF, in vitro fertilisation; IVF cases, pregnancies and children identified with conception assisted by IVF; Controls, pregnancies and children not identified as IVF assisted conception; ART, assisted reproductive technology; non-IVF ART, ovulation induction and intrauterine insemination; VPDC, Victorian Perinatal Data Collection; BDM, Victorian Births Death Marriages Registry; IVF, combined Victorian IVF pregnancy record database.

were included in the study (2,614 IVF cases and 100,184 controls were in both study arms). We estimate that our study cohort includes >95% of IVF conceptions during the study timeframe (Tables A and B in S3 File). Analysis of the linked and non-linked cases showed little evidence of association between linkage and exposure status (Chi$^2$ $p$ = 0.80); that is, IVF cases were just as likely to be included in the final linked cohort as controls. There were no births from 2014 that linked to outcome data.

Baseline population characteristics differed considerably between the 2 exposure groups (Table 2). Compared with spontaneously conceived controls, children conceived via IVF had older, more highly educated parents and mothers with lower parity. IVF-conceived children resided in postal areas with higher socioeconomic ranking and were less likely to be from non-English speaking backgrounds. Age at assessment was similar between the exposure groups.

## Global developmental vulnerability at school entry (The Australian Early Development Census, AEDC)

### Primary outcome

Our findings suggest no causal effect of IVF conception on developmental vulnerability, with 13.6% of IVF-conceived children predicted to be developmentally vulnerable (<10th percentile in 2 or more domains of the 5 AEDC domains) compared with 13.9% of spontaneously conceived children. The adjusted RD was at −0.3%, indicating that 0.3% fewer children who were conceived by IVF were developmentally vulnerable compared with those conceived spontaneously. However, the 95% CI (−3.7% to 3.1%), indicates this result is indistinguishable from zero. Similarly, the adjusted relative risk showed no detectable difference in risk of developmental vulnerability, where IVF-conceived children were 3.0% less likely to be developmentally vulnerable than spontaneously conceived children (RR 0.97, 95% CI: 0.77 to 1.25) (Table 3).

### Secondary outcomes

For secondary outcomes, we examined each of the 5 AEDC domains individually. The unadjusted observed results and causal model results for each individual domain are reported in Table 3. There were no differences between IVF- and spontaneously conceived children in adjusted risk difference for any of the individual AEDC domains.

### Missing data

Outcome data were missing for 5.6% of the AEDC-linked cohort. The vast majority (92%) of these missing cases were children with special needs (5.2% of overall cohort). There was no evidence of an association between the presence of missing outcome and exposure status (Chi$^2$ $p$ = 0.68). Sensitivity analysis was performed by (1) excluding children with special needs; and (2) including these children, with multiple imputation of their missing outcomes (Tables A and B in S8 File). Most covariates had minimal or no missing data (<1.0%). Maternal education level was missing for 30.5% and maternal post-school education was missing for 31.6%.

## Psychometric assessment of 5 educational domains at primary school (The National Assessment Program–Literacy and Numeracy, NAPLAN)

### Primary outcome

Our findings indicate the causal effect of IVF conception on overall NAPLAN z-score was indistinguishable from zero. The predicted outcome mean z-score and was 0.013 (SE 0.024)

**Table 3. Results of final causal model.**

| | Non-imputed crude data | | | Imputed data–causal model[a] | | | |
|---|---|---|---|---|---|---|---|
| | Proportions | | | Predicted proportions | | Regression co-efficient: | |
| | Controls (N = 168,503) | IVF (N = 4,697) | Unadjusted risk difference (95% CI)[b] | Controls (N = 168,503) | IVF (N = 1,506–1,560)[c] | ATE risk difference (95% CI) | ATE risk ratio (95% CI) |
| **Primary outcome (developmentally at risk in 2 or more domains)** | | | | | | | |
| | 0.140 | 0.104 | −0.036 (−0.045 to −0.027) | 0.139 | 0.136 | −0.003 (−0.037 to 0.031) | 0.97 (0.77 to 1.25) |
| **Secondary outcomes (developmentally at risk for individual domains)** | | | | | | | |
| Physical health and wellbeing | 0.126 | 0.105 | −0.021 (−0.030 to −0.012) | 0.125 | 0.124 | −0.0003 (−0.034 to 0.033) | 0.996 (0.76 to 1.31) |
| Social competence | 0.130 | 0.105 | −0.026 (−0.035 to −0.017) | 0.130 | 0.136 | 0.006 (−0.028 to 0.040) | 1.05 (0.80 to 1.37) |
| Emotional maturity | 0.125 | 0.103 | −0.022 (−0.031 to −0.013) | 0.125 | 0.114 | −0.011 (−0.042 to 0.021) | 0.91 (0.70 to 1.20) |
| Language and cognitive skills (school-based) | 0.107 | 0.076 | −0.031 (−0.039 to −0.023) | 0.106 | 0.118 | 0.012 (−0.020 to 0.045) | 1.12 (0.85 to 1.46) |
| Communication and general knowledge | 0.114 | 0.080 | −0.034 (−0.042 to −0.026) | 0.113 | 0.113 | 0.000 (−0.031 to −0.031) | 1.00 (0.76 to 1.31) |

Australian Early Development Census (children aged 4–6 years).

Children with missing outcome data identified as having special needs (5.2%) are included—their outcome category assumed to be "developmentally vulnerable."

[a] Causal model: multiply imputed data (imputation of covariates and outcomes) pooled estimates—doubly robust method: regression adjustment model with stabilised inverse probability weighting, plus trimmed for complete weight overlap. Variables included in model: sex at birth, age at assessment, language background other than English, socioeconomic status, maternal age, parity, and education.

[b] Small variation in case number for each of the 20 imputation datasets.

AEDC, Australian Early Development Census; ATE, Average treatment effect; CI, confidence interval; IVF, in vitro fertilisation (cases).

for IVF-conceived children and −0.016 (SE 0.002) for spontaneously conceived controls, with an adjusted mean difference of 0.030 (95% CI −0.018 to 0.077) (Table 4).

## Secondary outcomes

For secondary outcomes, we examined individual NAPLAN domain z-scores (Table 4). IVF-conceived children performed better on average in measures of writing than their spontaneously conceived peers with a z-score mean difference of 0.068 (95% CI 0.004 to 0.132), but this is unlikely to be a clinically important difference. The estimated effect is less than 0.07 of a standard deviation, and a difference of 0.2 standard deviations or greater was determined a priori as representing a finding of importance.

Additionally, for each domain, a binary outcome (domain scores above or below the national minimum standard) was examined. In 4 of 5 domains (numeracy, reading, spelling, and writing), IVF-conceived children were less likely to be below the national minimum standard compared with their spontaneously conceived peers (Table 5). For these 4 domains, the RD was between −0.7% and −1.25%. In absolute terms, this equates to approximately 1 additional IVF-conceived child, for every 100, predicted to score above the national minimum standard compared with their spontaneously conceived peers.

## Missing data

Spontaneously conceived children were more likely to have missing NAPLAN data (7.6%) than IVF-conceived children (5.9%, $Chi^2$ $p < 0.001$). During the primary analysis, missing

**Table 4. Results of final causal model.**

| | Non-imputed crude data | | | Imputed data–causal model[a] | | |
|---|---|---|---|---|---|---|
| | Mean (SE) | | | Potential outcome mean (SE) | | Regression co-efficient: |
| | Control (N = 333,335) | IVF (N = 8,976) | Unadjusted mean difference (95% CI)[b] | Controls (N = 333,335) | IVF (N = 1,985–2,010)[c] | ATE mean difference (95% CI) |
| **Primary outcome—overall Z-score** | | | | | | |
| | −0.006 (0.002) | 0.232 (0.011) | 0.238 (0.217 to 0.260) | −0.016 (0.002) | 0.013 (0.024) | 0.030 (−0.018 to 0.077) |
| **Secondary outcomes–individual domains Z-score** | | | | | | |
| Grammar and punctuation | −0.107 (0.002) | 0.209 (0.015) | 0.316 (0.286 to 0.345) | −0.113 (0.003) | −0.085 (0.033) | 0.028 (−0.036 to 0.092) |
| Numeracy | −0.113 (0.002) | 0.174 (0.015) | 0.286 (0.258 to 0.316) | −0.118 (0.003) | −0.070 (0.033) | 0.048 (−0.018 to 0.113) |
| Reading | −0.108 (0.002) | 0.226 (0.015) | 0.334 (0.305 to 0.263) | −0.113 (0.003) | −0.092 (0.033) | 0.021 (−0.042 to 0.085) |
| Spelling | −0.085 (0.002) | 0.123 (0.012) | 0.208 (0.184 to 0.231) | −0.091 (0.002) | −0.083 (0.029) | 0.008 (−0.048 to 0.064) |
| Writing | −0.115 (0.002) | 0.137 (0.014) | 0.252 (0.223 to 0.280) | −0.124 (0.002) | −0.056 (0.033) | 0.068 (0.004 to 0.132)[d] |

Grade 3 (fourth year of schooling) NAPLAN z-scores (children aged 7–9 years).

Children with missing outcome data exempt from NAPLAN (2.1%) are included—their z-score set at lowest possible score.

[a] Causal model: imputed data pooled estimates—regression adjustment model with stabilised inverse probability weighting, trimmed for complete weight overlap.

Variables included in model: sex at birth, age at assessment, language background other than English, socioeconomic status, maternal age, parity, education, and second parent education.

[b] Small variation in case number for each of the 20 imputation datasets.

[c] 95% confidence interval does not cross the null.

[d] Linear regression/unpaired *t* test.

ATE, average treatment effect; IVF, in vitro fertilisation (cases); NAPLAN, National Assessment Program for Literacy and Numeracy; SE, standard error.

outcomes related to a child being absent or withdrawing from the test were imputed. The results presented include 7,222 children who were exempt from sitting the NAPLAN, with their results set to the lowest possible outcome score. Sensitivity analysis was performed by (1) excluding these children; and (2) including the exempt cases, with multiple imputation of their missing outcomes. There was no meaningful difference in the results (Tables A and B in S9 File). Most covariates had minimal or no missing data (<4.0%). Second parent school education level was missing in 13.8% of cases and post-school education missing in 15.4% of cases.

## Sensitivity analyses

To validate our analysis model, we re-examined our AEDC primary outcome and the NAPLAN binary domain outcomes using TMLE modelling. Results from the TMLE model did not meaningfully differ from the findings of the primary analysis (Table A in S10 File).

An E-value was estimated for both primary outcomes and was found to be 1.90 and 1.77 for AEDC and NAPLAN outcomes, respectively, suggesting that an unknown bias of sufficient magnitude to change the study findings is unlikely (Figs A and B in S11 File).

## Discussion

Using a causal inference approach, we found no effect of IVF conception on developmental vulnerability at school entry in Victorian children born between 2005 and 2014. Additionally, IVF-conceived children performed as well as their spontaneously conceived peers in school-based psychometric testing at age 7 to 9 years.

For the first time, our study has estimated the causal effect of IVF conception on global childhood development at school entry and educational outcomes at primary school, under the assumptions of causal inference. Using an updated epidemiological approach [25], this

**Table 5. Results of final causal model.**

| | Non-imputed crude data | | | Imputed data–causal model[a] | | | |
|---|---|---|---|---|---|---|---|
| | Proportions | | | Predicted outcome proportions | | Regression co-efficient: | |
| | Control (N = 333,335) | IVF (N = 8,976) | Unadjusted risk difference (95% CI)[b] | Control (N = 333,335) | IVF (N = 1,985–2,010)[c] | ATE risk difference (95% CI) | ATE risk ratio (95% CI) |
| Secondary outcomes–individual domains below "National Minimum Standard" | | | | | | | |
| Grammar and punctuation | 0.051 | 0.029 | −0.022 (−0.026 to −0.018) | 0.052 | 0.044 | −0.008 (−0.020 to 0.003) | 0.84 (0.57 to 1.24) |
| Numeracy | 0.038 | 0.024 | -0.015 (-0.018 to -0.011) | 0.039 | 0.028 | −0.011 (−0.021 to −0.002)[d] | 0.72 (0.48 to 1.06) |
| Reading | 0.044 | 0.026 | −0.018 (−0.022 to −0.015) | 0.046 | 0.035 | −0.011 (−0.022 to −0.001)[d] | 0.75 (0.51 to 1.12) |
| Spelling | 0.049 | 0.028 | −0.022 (−0.025 to −0.018) | 0.051 | 0.039 | −0.012 (−0.023 to −0.001)[d] | 0.77 (0.52 to 1.14) |
| Writing | 0.033 | 0.020 | −0.012 (−0.015 to −0.009) | 0.034 | 0.021 | −0.012 (−0.021 to −0.004)[d] | 0.63 (0.43 to 0.94)[d] |

Grade 3 (fourth year of schooling) NAPLAN, scores below "National Minimum Standard."

Children with missing outcome data exempt from NAPLAN (2.1%) are included—their outcome is "below national minimum standard."

[a] Causal model: imputed data pooled estimates—regression adjustment model with stabilised inverse probability weighting, trimmed for complete weight overlap.

Variables included in model: sex at birth, age at assessment, language background other than English, socioeconomic status, maternal age, parity, education, and second parent education.

[b] Small variation in case number for each of the 20 imputation datasets.

[c] 95% confidence interval does not cross the null.

[d] Chi$^2$ test.

ATE, average treatment effect; IVF, in vitro fertilisation (cases); NAPLAN, National Assessment Program for Literacy and Numeracy; SE, standard error.

study provides robust evidence about the longer term implications of IVF conception. The findings of this study offer timely reassurance about the impact of IVF conception on the developmental and educational outcomes at primary school age of the children conceived. Neither the outcomes of developmental vulnerability at school entry nor educational achievement at age 7 to 9 differed substantially between IVF- and spontaneously conceived children. Among 4 out of 5 NAPLAN individual domain national minimum standard results, there was a trend towards better performance in the IVF cases, but the clinical and social implications of these findings are difficult to quantify.

Two large Scandinavian studies have reported on childhood outcomes following IVF conception. Norrman and colleagues found that IVF-conceived children perform worse on school-based assessment in year 9 [18], among their cohort of just over 8,000 IVF-conceived children. Wienecke and colleagues reported that IVF-conceived children had poorer school performance than controls and that spontaneously conceived children of subfertile parents also had poorer outcomes [19]. By examining a subpopulation of spontaneously conceived children of subfertile parents, the authors of this Danish study concluded that the IVF process itself was not responsible for the differences demonstrated [19].

These past studies are limited by examining historical birth cohorts dating back prior to the year 2001. Our study examines a more contemporary birth cohort (2005 to 2014), which is important given the advances in artificial reproductive techniques that have occurred since the

turn of the century. IVF technologies that have evolved since this time include the introduction of blastocyst culture, vitrification, and single-embryo transfer [4,43]. Thus, our study findings are more generalisable to contemporary fertility practice. Importantly, our use of updated epidemiological and statistical methods ensures that we have estimated effects that have a causal interpretation. It is important that our methods are replicated in future studies to strengthen the existing evidence base.

Given the use of observational data, there were missing data and inherent differences in the covariate profile of the exposure cohorts. An a priori SAP was developed to overcome these limitations. First, inverse probability weighting with regression adjustment was used to mimic exchangeable treatment and control comparison groups, similar to those that would be generated by randomisation in a controlled trial. The success of this procedure is demonstrated by achieving adequate covariate balance and thus sufficient overlap of covariate distributions between exposure groups after inverse probability weighting (Figs A–D in S6 File). Second, we sought to mitigate the potential biases resulting from missing data. In order to do this, we performed multiple imputation of covariates included in our model and then compared the results of analyses that were based on complete cases with those of multiply imputed datasets (Tables A and B in S12 File).

It is possible that unmeasured common cause confounders may have led to bias in estimating the ATEs. Many important factors (socioeconomic status, maternal age, and education) were identified a priori, measured, and included in the estimation procedure. Potential known but unmeasured sources of bias include subfertility and maternal BMI. Current evidence suggests that subfertility is likely to be associated with poorer childhood outcomes [19]. Consequently, if this variable were able to be measured and included in our causal model, correcting for it is likely to have favoured IVF-conceived children in our analysis. Maternal BMI is also likely to have followed the same trend with higher average BMI among the IVF group (after accounting for socioeconomic position) and high BMI being associated with poorer perinatal and childhood outcomes [44].

Unmeasured variables may have had an impact on the outcome. Factors such as childcare attendance or grandparent involvement will be preceded on causal pathways by covariates that were measured and included in the model, such as maternal age and socioeconomic status [45–47]. These factors were therefore considered to mediate rather than confound the relationship between these covariates and the outcome. Sensitivity analyses were performed to further evaluate unmeasured confounding, with E-values calculated for AEDC and NAPLAN primary outcomes. Within the limitation of E-values, these analyses indicate that it is unlikely an unknown bias exists without our knowledge and with the necessary magnitude of effect and prevalence to change our conclusions (Figs A and B in S11 File) [48].

Generalisation of our findings to all IVF births is a potential study limitation. As described in our Methods, observations with non-overlapping PSs were excluded from analysis in order to meet the assumption of positivity, required for causal inference under the potential outcomes framework. Generalisation of our findings to all IVF births therefore requires the consideration that the baseline characteristics of the population of interest are comparable to the IVF cases analysed in our final cohort.

Due to the use of school-based outcome assessments, our cohort was limited to children attending school. AEDC, as a triennial assessment, limited our sample to children captured during assessment years, and the later years of our birth cohort had not yet reached the assessment age for NAPLAN outcomes to be captured. However, our study included 70% of the relevant birth cohort for the study timeframe and in the years where both AEDC and NAPLAN data were available, over 95% of the Victorian birth cohort was sampled (Table A in S3 File). The remaining approximately 5% of children not sampled represent failed linkages as well as

excluded IVF conceptions (due to non-Victorian IVF or non-IVF-assisted reproduction). A small percentage will also represent children with a disability significant enough not to attend mainstream school, introducing potential selection bias. Importantly, however, our study was not designed to assess severe disability or developmental delay, but rather an overall measure of global development and school achievement.

Furthermore, through the examination of school-based outcomes, our study was inherently designed to examine outcomes for liveborn children. "Live birth bias" as it is known, is a recognised limitation of observational studies that investigate periconception and antenatal exposures [49]. For the purposes of this research question, the outcomes of failed conception, miscarriage, stillbirth are considered alternative endpoints and less relevant to the research question that aims to compare the school-age outcomes of children born following IVF conception with those who were conceived without assistance.

Under the specified assumptions, this analysis has demonstrated that there is no causal effect within the population studied of IVF conception on early childhood developmental vulnerability and school-age educational outcomes. Compared with spontaneously conceived children, children conceived by IVF were no more likely to be developmentally vulnerable at school entry and had equivalent numeracy and literacy performance by age 7 to 9 years. These findings provide important reassurance for current and prospective parents and their treating clinicians.

## Supporting information

**S1 Fig. Direct acyclic graph.**
(DOCX)

**S1 File. Methods: Description of outcome metrics.**
(DOCX)

**S2 File. Study protocol.**
(DOCX)

**S3 File.** Tables A and B. Table A. Successful linkages by birth year. Table B. Annual cycle summaries from major Victorian IVF providers 2010–2014.
(DOCX)

**S4 File.** Fig A. Analysis flow chart (NAPLAN).
(DOCX)

**S5 File. Methods: Multiple imputation model summary and model diagnostics.** Table A. Missing data summary (NAPLAN). Fig A. NAPLAN convergence. Fig B. NAPLAN density plots of observed and imputed data. Fig C. NAPLAN distribution of outcome and covariates after imputation in m = 1 dataset.
(DOCX)

**S6 File.** Figs A–D: Distribution and overlap of manually calculated stabilised weights. Fig A. AEDC imputation #1. Fig B. AEDC imputation #13. Fig C. NAPLAN imputation #1. Fig D. NAPLAN imputation #13.
(DOCX)

**S7 File.** Figs A–C: Variable standardised mean differences. Fig A. NAPLAN imputation #1 variable standardised mean differences. Fig B. NAPLAN imputation #13 variable standardised mean differences. Fig C. AEDC imputation #7 variable standardised mean differences.
(DOCX)

**S8 File.** Tables A and B. Table A. Sensitivity analysis–AEDC (special needs multiply imputed). Table B. Sensitivity analysis–AEDC (special needs excluded).
(DOCX)

**S9 File.** Tables A and B. Table A. Sensitivity analysis–NAPLAN (exempt multiply imputed). Table B. Sensitivity analysis–NAPLAN (exempt excluded).
(DOCX)

**S10 File.** Table A–Sensitivity analysis–binary outcomes TMLE.
(DOCX)

**S11 File.** Figs A and B. Fig A. Sensitivity analysis AEDC primary outcome–E-value estimation. Fig B. Sensitivity analysis NAPLAN primary outcome–E-value estimation.
(DOCX)

**S12 File.** Tables A and B. Table A. Traditional regression and treatment effect models (AEDC). Table B. Traditional regression and treatment effect models (NAPLAN).
(DOCX)

**S13 File. STROBE guideline checklist.**
(DOCX)

## Acknowledgments

This paper uses data from the Australian Early Development Census (AEDC). The AEDC is funded by the Australian Government Department of Education, Skills and Employment. The findings and views reported are those of the author(s) and should not be attributed to the Department or the Australian Government. We are grateful for the provision of data by the AEDC.

We are grateful to CCOPMM for providing access to the data used for this project and for the assistance of the staff at Safer Care Victoria. The conclusions, findings, opinions and views, or recommendations expressed in this paper are strictly those of the author(s). They do not necessarily reflect those of CCOPMM.

We are thankful for contribution of Victorian IVF providers, Melbourne IVF, Monash IVF, and City Fertility Centre to this research. We acknowledge the significant amount of work undertaken on behalf of this project and appreciate the opportunity to work with staff from each unit.

Finally, we are grateful to the Australian Curriculum Assessment and Reporting Authority (ACARA) for their assistance, collaboration, and for providing the National Assessment Program for Literacy and Numeracy (NAPLAN) data.

## Author Contributions

**Conceptualization:** Amber L. Kennedy, Beverley J. Vollenhoven, Richard J. Hiscock, Catharyn J. Stern, Susan P. Walker, Jeanie L. Y. Cheong, Jon L. Quach, Roxanne Hastie, David Wilkinson, John McBain, Lyle C. Gurrin, Stephen Tong, Anthea C. Lindquist.

**Data curation:** Amber L. Kennedy, Beverley J. Vollenhoven, Richard J. Hiscock, David Wilkinson, Vivien MacLachlan, Franca Agresta, Susan P. Baohm, Anthea C. Lindquist.

**Formal analysis:** Amber L. Kennedy, Richard J. Hiscock, Jon L. Quach, Lyle C. Gurrin, Anthea C. Lindquist.

**Funding acquisition:** Amber L. Kennedy, Beverley J. Vollenhoven, Catharyn J. Stern, Susan P. Walker, David Wilkinson, John McBain, Anthea C. Lindquist.

**Investigation:** Amber L. Kennedy, Richard J. Hiscock, Catharyn J. Stern, Susan P. Walker, Jon L. Quach, Lyle C. Gurrin, Anthea C. Lindquist.

**Methodology:** Amber L. Kennedy, Beverley J. Vollenhoven, Richard J. Hiscock, Catharyn J. Stern, Susan P. Walker, Jeanie L. Y. Cheong, Jon L. Quach, Roxanne Hastie, Lyle C. Gurrin, Stephen Tong, Anthea C. Lindquist.

**Project administration:** Amber L. Kennedy, Beverley J. Vollenhoven, Catharyn J. Stern, Susan P. Walker, Jeanie L. Y. Cheong, Jon L. Quach, David Wilkinson, John McBain, Vivien MacLachlan, Franca Agresta, Susan P. Baohm, Anthea C. Lindquist.

**Resources:** Amber L. Kennedy, Beverley J. Vollenhoven, Catharyn J. Stern, Susan P. Walker, Jon L. Quach, David Wilkinson, John McBain, Stephen Tong, Anthea C. Lindquist.

**Software:** Amber L. Kennedy, Richard J. Hiscock, Anthea C. Lindquist.

**Supervision:** Beverley J. Vollenhoven, Richard J. Hiscock, Catharyn J. Stern, Susan P. Walker, Lyle C. Gurrin, Stephen Tong, Anthea C. Lindquist.

**Validation:** Richard J. Hiscock, Jon L. Quach, Lyle C. Gurrin, Stephen Tong, Anthea C. Lindquist.

**Visualization:** Richard J. Hiscock, Anthea C. Lindquist.

**Writing – original draft:** Amber L. Kennedy, Richard J. Hiscock, Anthea C. Lindquist.

**Writing – review & editing:** Amber L. Kennedy, Beverley J. Vollenhoven, Richard J. Hiscock, Catharyn J. Stern, Susan P. Walker, Jeanie L. Y. Cheong, Jon L. Quach, Roxanne Hastie, David Wilkinson, John McBain, Lyle C. Gurrin, Vivien MacLachlan, Franca Agresta, Susan P. Baohm, Stephen Tong, Anthea C. Lindquist.

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
