## [Editor Report · Decision Letter 0]

4 Jul 2022

Dear Dr Lindquist, 

Thank you for submitting your manuscript entitled "School-age outcomes among IVF-conceived children: a causal inference analysis using linked population-wide data" for consideration by PLOS Medicine.

Your manuscript has now been evaluated by the PLOS Medicine editorial staff as well as by an academic editor with relevant expertise and I am writing to let you know that we would like to send your submission out for external peer review.

Please re-submit your manuscript within two working days, i.e. by Jul 06 2022 11:59PM.

Kind regards,

Philippa

Dr Philippa C Dodd, MBBS MRCP PhD

Senior Editor

PLOS Medicine

---

## [Decision Letter · Decision Letter 1]

6 Sep 2022

Dear Dr. Lindquist,

Thank you very much for submitting your manuscript "School-age outcomes among IVF-conceived children: a causal inference analysis using linked population-wide data" (PMEDICINE-D-22-02257R1) for consideration at PLOS Medicine. 

[LINK]

In light of these reviews, I am afraid that we will not be able to accept the manuscript for publication in the journal in its current form, but we would like to consider a revised version that addresses the reviewers' and editors' comments. Obviously we cannot make any decision about publication until we have seen the revised manuscript and your response, and we plan to seek re-review by one or more of the reviewers. 

We expect to receive your revised manuscript by Sep 27 2022 11:59PM. Please email us (plosmedicine@plos.org) if you have any questions or concerns.

We look forward to receiving your revised manuscript. 

Sincerely,

Philippa Dodd, MBBS MRCP PhD

PLOS Medicine

plosmedicine.org

ABSTRACT

Abstract Methods and Findings: 

Please quantify the main results with 95% CIs and p values.

Please include the important dependent variables that are adjusted for in the analyses.

In the last sentence of the Abstract Methods and Findings section, please describe the main limitation(s) of the study's methodology.

INTRODUCTION

Please conclude the Introduction with a clear description of the study question or hypothesis.

METHODS and RESULTS

Please provide 95% CIs and p values where relevant 

When a p value is given, please specify the statistical test used to determine it.

GENERAL

in the context of the reviewer comments below, it might be necessary to revise your title

Please do so according to PLOS Medicine's style. Your title must be nondeclarative and not a question. It should begin with main concept if possible. "Effect of" should be used only if causality can be inferred, i.e., for an RCT. Please place the study design ("A randomized controlled trial," "A retrospective study," "A modelling study," etc.) in the subtitle (ie, after a colon) 

Please ensure that the study is reported according to the STROBE guideline and include the completed STROBE checklist as Supporting Information. Please add the following statement, or similar, to the Methods: "This study is reported as per the Strengthening the Reporting of Observational Studies in Epidemiology (STROBE) guideline (S1 Checklist)." The STROBE guideline can be found here: http://www.equator-network.org/reporting-guidelines/strobe/ When completing the checklist, please use section and paragraph numbers, rather than page numbers.

Thank you for including a prospective analysis plan, please include any changes made to the analyses-- including those made in response to peer review comments-- in the Methods section of the paper, with rationale.

In the manuscript text, please indicate: (1) the specific hypotheses you intended to test, (2) the analytical methods by which you planned to test them, (3) the analyses you actually performed, and (4) when reported analyses differ from those that were planned, transparent explanations for differences that affect the reliability of the study's results. If a reported analysis was performed based on an interesting but unanticipated pattern in the data, please be clear that the analysis was data-driven.

DISCUSSION

Please remove sub-headings such that the discussion is presented follows: a short, clear summary of the article's findings; what the study adds to existing research and where and why the results may differ from previous research; strengths and limitations of the study; implications and next steps for research, clinical practice, and/or public policy; one-paragraph conclusion.

Comments from the academic editor:

1. I do not buy the causal language. All observational studies have potential for unmeasured confounders. I think the use and emphasis of causal language could actually distract attention from the main message, which I think is important. In fact, the unadjusted data indicate an apparent protective effect of IVF which disappears on adjustment (Table 2). It would be good to add 95% CI to the unadjusted analysis to see if this was beyond the play of chance. But it is apparent that measured parental characteristics are associated with an increased chance of using IVF and a decreased risk of poor educational outcome. Hence, it follows that unmeasured parental characteristics associated with an increased chance of using IVF and a decreased risk of poor educational outcome could be masking an adverse effect of IVF and the use of causal language is wrong.

2. It would be good to have a positive control, i.e. data on a factor which we know is associated with educational outcome. That way we know their methods can detect an association if there is one there. Preterm birth is an obvious one. Perhaps an easy way to do this is to provide coefficients for all the covariables that they included in adjustment.

3. It would actually be quite interesting to see what the results are if they applied a more commonly used method of multivariate adjustment, such as logistic regression or propensity score analysis just to see how different (if at all) the results are with the method they employ. Comments 2 and 3 could be addressed by providing a supplementary table of unadjusted and adjusted ORs from logistic regression to allow comparison.

4. I agree with the comments of the statistician. But it is reassuring that their overall assessment is positive.

Comments from the reviewers:

Reviewer #1 (methodological reviewer): This aim of this study was to determine the causal effect of in-vitro fertilization (IVF) conception on primary schoolage childhood developmental and educational outcomes, compared with outcomes following spontaneous conception.

Comments:

"Covariates to be considered for inclusion in the statistical analysis models were decided a priori by the authorship team whose expertise included epidemiology, perinatology, reproductive endocrinology and education"

The authors have satisfactorily adjusted for potential confounding in the analysis.

"For the pre-specified statistical analysis plan (SAP) and the directed acyclic graph (DAG), see supplementary file (eFigure 1 in S3)."

The authors have suitably provided the DAG in the supplementary material.

Can the authors also please provide a copy of the SAP in the supplementary material (it is not currently clear if or where this is attached to the file)?

"A doubly robust inverse-probability-weighted regression adjustment (IPWRA) model43,44 was used..."

The authors have applied technically appropriate and rigorous statistical methods within the context of this research.

"Provided assumptions are satisfied, the estimates generated from these routines can be interpreted as the population average causal effect"

Can the authors please include a transparent and thorough discussion on the assumptions that are required for causal inferences to be drawn?

"Analysis involved detailed examination of missing data, consideration of missing outcome data, multiple imputation of missing covariate data, consideration of clustering within mothers and adjustments to effect size modelling. Finally, sensitivity analysis was also performed to address identified sources of potential bias."

Can the authors please provide more detail on these important analyses within the main article here?

"Overall, a total of 11,059 IVF-conceived children and 401,654 spontaneously conceived children were included in the study (2,614 IVF cases and 100,184 controls were in both study arms)."

Whilst the authors have examined that "Analysis of the linked and nonlinked cases showed little evidence of association between linkage and exposure status (p = 0.80); that is, IVF cases were just as likely to be included in the final linked cohort as controls", can the authors please further comment on whether the final included samples analysed in this study can be considered to be representative of the wider populations of interest within the context of this research?

"Outcome data were missing for 5.6% of the AEDC-linked cohort. The vast majority (92%) of these missing cases were children with special needs (5.2% of overall cohort). There was no evidence of an association between the presence of missing outcome and exposure status (p = 0.68). Sensitivity analysis was performed by 1) excluding children with special needs and 2) including these children, with multiple imputation of their missing outcomes (see supplementary file: eTable 4 and 5 - Sensitivity Analysis (AEDC) in S11 and s12). "

and

"Spontaneously conceived children were more likely to have missing NAPLAN data (7.6%) than IVF-conceived children (5.9%, p<0.001). During the primary analysis, missing outcomes related to a child being absent or withdrawing from the test, were imputed. The results presented include 7,222 children who were exempt from sitting the NAPLAN, with their results set to the lowest possible outcome score. Sensitivity analysis was performed by 1) excluding these children and 2) including the exempt cases, with multiple imputation of their missing outcomes. There was no meaningful difference in the results (see supplementary file: eTable 6 and 7 - Sensitivity Analysis (NAPLAN) in S13 and S14). "

The authors have appropriately communicated and handled missing outcome data within the analysis.

"Maternal education level was missing for 30.5% and maternal post-school education was missing for 31.6%."

and

"Second parent school education level was missing in 13.8% of cases and post-school education missing in 15.4% of cases. "

Whilst the authors discuss missing data in the discussion of the study limitations, can the authors please clarify and expand on how they dealt with missing covariate data in the Methods and Results sections?

Overall, the results are presented clearly and the main study limitations have been thoroughly addressed.

Reviewer #2: The authors report the lack of an association of IVF with later childhood educational outcomes using routinely collected data from Australia. This is an extremely well conducted and well written study, with robust and appropriate sensitivity analyses which adds substantially to the field and will reassures many parents. 

Minor comments:

The authors make a big deal of the causal analysis, and I would suggest statistical review to confirm the appropriateness of the methodology and causal claims. I am conscious that the same used to be said regarding matching and that has now shown to be incorrect. I was not convicned that teh paper would materially suffer by removing the "casual anlaysis" claims in the title or intorudction etc. 

I wondered if graphical representation of the results would further enhance the paper. 

Reviewer #3: This study examined associations of ART conception with School-age outcomes using register data from Australia. The authors should be commended for trying to provide more reliable causal evidence from observational data, something we should all be doing. I have only a few minor comments

eTable 1 should be brought into the main paper and / or a separate section in the methods should be added on the target trial emulation procedure

Where is the study protocol published / date it was published?

In line with the glossary, the exposure of interest should be referred to as ART rather than IVF

The aim of the study was to emulate a target trial however, baseline characteristics differed considerably between the treatment groups. Can you describe the implications of this. Does it mean that trial emulation failed? Is there a way you can try to fix this?

If data are available, it would be of interest to present results separately for fresh and frozen embryo transfers. Please consider this.

[LINK]

---

## [Decision Letter · Decision Letter 2]

18 Oct 2022

Dear Dr. Lindquist,

Thank you very much for re-submitting your manuscript "School-age outcomes among IVF-conceived children: a causal inference analysis using linked population-wide data" (PMEDICINE-D-22-02257R2) for review by PLOS Medicine.

I have discussed the paper with my colleagues and the academic editor and it was also seen again by 4 reviewers. I am pleased to say that provided the remaining editorial and production issues are dealt with we are planning to accept the paper for publication in the journal.

[LINK]

We look forward to receiving the revised manuscript by Oct 25 2022 11:59PM.   

Sincerely,

Philippa Dodd, MBBS MRCP PhD

Senior Editor 

PLOS Medicine

plosmedicine.org

Requests from Editors:

GENERAL

Please address all editor and reviewer comments detailed below, in full

Please remove the funding/financial disclosure/

DATA AVAILABILITY STATEMENT

Please provide a URL for The Centre for Victorian Data Linkage

AUTHOR SUMMARY

Thank you for including an author summary. 

Line 74: Consider an alternative term to “defect” perhaps “congenital anomalies” in place of “…birth defects, congenital abnormalities…” to account for both structural and functional anomalies, or something similar

REFERENCES

For in-text reference call-outs, citations should be placed within square brackets and preceding punctuation, as follows: “…asymptomatically [2,4].” 

Please check to ensure that the bibliography is listed in line with our guidance which can be found here: https://journals.plos.org/plosmedicine/s/submission-guidelines#loc-references

TABLE 1

To improve accessibility to the reader, please adjust row width/text spacing to ensure IQRs are reported in a single row or place below the relevant data point

SUPPORTING INFORMATION

Please ensure all figures and tables in the supporting files have abbreviations defined

S2 TABLE 1 - Please define IVF, ITT, AEDC, NAPLAN

S4 TABLE 2b – please define “#" and "ICSI"

S5 eMETHODS - please ensure referencing Is formatted as detailed above including in-text reference call-outs

S6 FIGURE 2 - please define NAPLAN - please check and amend throughout all supplementary files 

S10 ALL FIGURES - please define the abbreviations - LBOTE, SEIFA, NAPLAN

S16 - please define AEDC, NAPLAN, DV2

S17 TABLE 9 and 10 - please define AEDC and NAPLAN

SOCIAL MEDIA

To help us extend the reach of your research, please provide any Twitter handle(s) that would be appropriate to tag, including your own, your coauthors’, your institution, funder, or lab. Please respond to this email with any handles you wish to be included when we tweet this paper.

Comments from Reviewers:

Reviewer #1: Many thanks to the authors for satisfactorily considering and responding to each comment in turn, suitably amending the manuscript as requested.

Reviewer #2: I would stand by my comments that removal of the causal claim in the title would improve the paper, I am not a fan of playing different referees of each other several of us have highlighted that the causal language may be overstated. 

"School-age outcomes among IVF-conceived children" is snappier and cuts to the chase.

Reviewer #4: I am grateful for the opportunity to review this interesting and innovative study that seeks to estimate the total causal effect of conception by assisted reproductive technology on educational outcomes in school-age children. I was approached to provide a methodological review of the appropriateness and accuracy of the causal inference methods as well as the reporting and use of language. My review will focus primarily on these points.

Overall, I believe the study is has been well conducted and the research team should be commended for their hard work and diligence. I believe the research is scientifically sound but I believe the quality of the manuscript could be improved with some language and presentational changes as detailed below.

1) TITLE: 'a causal inference analysis' is a vague term. I think it would be better for the study to be described more precisely as something like, 'a target trial emulation study using augmented inverse probability of treatment weighting'.

2) INTRODUCTION, METHODS, AND RESULTS: A causal inference approach encourages a focus on interval estimation rather than (null) hypothesis testing. I therefore find it somewhat strange that the study states a null hypothesis at the end of the introduction and goes on to report null-distribution p-values. I think it would be less jarring if the authors instead described their aim/s as something more like 'to estimate the total causal effect of conception by IVF on the risk of developmental vulnerability'. Similar language would apply in the methods and results, e.g. rather than saying that 'The null hypothesis of no causal effect of IVF conception on developmental vulnerability was supported by our findings…' the study would simply report something like 'the estimated total causal effect of conception by IVF on the risk of developmental vulnerability was indistinguishable from zero (risk difference = -0.3%, 95% CI: -3.7% to 3.1%)'. 

3) INTRODUCTION (lines 127-140): Although a target trial emulation study is indeed designed to 'mimic' a randomized controlled trial, I think this wording is likely to raise unfortunate associations among readers familiar with poorly-conducted propensity score analyses as 'mimicking randomised controlled trials'. I would therefore recommend toning down the language a little and simply stating that the study aimed to emulate a target trial by using the method introduced by Hernan et al with the augmented inverse probability of treatment weighting estimator. Similarly, I don't particularly like the language of achieving 'balance' in the distribution of baseline characteristic, in part because this is not what propensity score methods do (they aim to balance the outcome propensities, not the covariates). Comments about how this is 'similar to what occurs when participants are randomised' may be theoretically true, but this is unlikely to ever be achieved in practice. That said, I commend the authors for being clear about their causal aims. As outlined by Haber et al 2022 (Causal and Associational Language in Observational Health Research: A Systematic Evaluation) it is important for researchers to be clear where they seek to estimate a causal effect. It is just not necessary to 'over egg' this ambition or the methods used to achieve this aim. Thus, it is good to state an aim to 'estimate the total causal effect of conception by IVF on developmental vulnerability', and interpret the results as 'estimates of' this total causal effect, but perhaps not so good to start talking about 'mimicking randomised controlled trials'.

4) METHODS: I would prefer the target trial emulation table to be presented in the main manuscript. I believe Plos Medicine is an online-only journal, so I see no reason why this should not be possible. 

5) METHODS: There is a gap between the allocation and completion of the treatment because the conception must survive until live birth to be counted as belonging to either exposure group. In the target trial, what would happen to the pregnancies that don't result in live birth? I expect there will be an unequal chance of surviving to live birth between the in-vivo and in-vitro conceptions, creating the chance of selection bias. This ought to be acknowledged as a potential source of bias or, ideally, accounted for as competing events. 

6) METHODS: In terms of exclusion criteria, what happened to the twins and other multiple pregnancies? I cannot see them described in the exclusion criteria. 

7) METHODS: Regarding the target trial table, I think the authors could be more explicit in both the target trial and (in particular) the emulation study columns. I know it seems strange, but where the details of the emulation study are identical to the target trial, this really ought to be explicitly stated (either by repeating the details or simply saying 'same as for the target trial'). For example, if the inclusion criteria for the trial are 'all couples wanting to conceive and weight the ability to conceive' then the emulation column should repeat the same. In general, the emulation column does not really seem to state the expected criteria. E.g. the assignment procedures box should summarise your method for achieving conditional exchangeability and your causal contrasts box should start the target estimands. The recent paper by Matthews et al in the BMJ (Target trial emulation: applying principles of randomised trials to observational studies) offers a little help here about what should be mentioned in the table.

8) METHODS: I think that the supplementary methods should be included in the main manuscript, which is currently rather light on details.

9) METHODS, RESULTS, and throughout: Since this study uses a DAG, I believe it would benefit from following the reporting recommendations outlined in Tennant et al 2021 (Use of directed acyclic graphs (DAGs) to identify confounders in applied health research: review and recommendations). 

10) METHODS and RESULTS: The authors present 'crude' associations in the non-weighted sample. I believe these results may have been added on the instruction of a previous reviewer, as per traditional. Alas, these associations should not be reported as they are likely to be extremely biased and prone to misinterpretation. Only the best estimate of the target estimands should be reported in the main manuscript. All sub-analyses involving alternative estimators (whether unweighted logistic regression, TLME, or - if absolutely necessary - the unweighted and unconditional analyses) should be reserved for the supplementary materials. 

11) RESULTS: Null hypothesis significance testing is strongly discouraged in epidemiological analyses. The null-distribution p-values are therefore unnecessary and do not need to be reported. The authors may wish to present their E-values alongside the point estimates and confidence intervals instead.

[LINK]

---

## [Decision Letter · Decision Letter 3]

14 Nov 2022

Dear Dr. Lindquist,

Thank you very much for re-submitting your manuscript "School-age outcomes among IVF-conceived children." (PMEDICINE-D-22-02257R3) for review by PLOS Medicine.

I have discussed the paper with my colleagues and the academic editor and it was also seen again by 1 reviewer. I am pleased to say that provided the remaining editorial and production issues are dealt with we are planning to accept the paper for publication in the journal.

[LINK]

We look forward to receiving the revised manuscript by Nov 17 2022 11:59PM.   

Sincerely,

Pippa

Philippa Dodd, MBBS MRCP PhD

PLOS Medicine

plosmedicine.org

Requests from Editors:

1) Thank you for revising your title in context of previous reviewer comments. With these in mind, please revise your title according to PLOS Medicine's style. Your title must be nondeclarative and not a question. It should begin with main concept if possible. "Effect of" should be used only if causality can be inferred, i.e., for an RCT. Please place the study design ("A randomized controlled trial," "A retrospective study," "A modelling study," etc.) in the subtitle (ie, after a colon).

2) Where you report adjusted analyses in data tables 3, 4 and 5 of the main manuscript, please also include the unadjusted analyses, either within an additional column in the existing tables or as additional supplementary files.

3) Thank you for updating your data availability statement. Please ensure that the URL is placed in the manuscript submission form when you re-submit your manuscript.

4) Thank you for including twitter handles. Please also ensure that these are placed in the manuscript submission form when you re-submit your manuscript.

5) Please remove the COI and data availability statements from the end of the main manuscript and include only in the manuscript submission form

6) Throughout, please replace "Fig" with "Figure", including in the supplementary files

7) Please define DV2 in supplementary figures 13 and 14 - please check throughout and ensure all abbreviations are clearly defined within relevant capitions.

Comments from the Academic Editor:

The authors have done a nice job with revisions and happy to move to accept

[LINK]

---

## [Editor Report · Decision Letter 4]

23 Nov 2022

Dear Dr Lindquist, 

On behalf of my colleagues and the Academic Editor, Dr Sarah Stock, I am pleased to inform you that we have agreed to publish your manuscript "School-age outcomes among IVF-conceived children: a population-wide cohort study" (PMEDICINE-D-22-02257R4) in PLOS Medicine.

Before your manuscript can be published you will need to address the following which we asked for but could not locate:

1) The completed STROBE checklist was not available in this version of your manuscript, please include AND

2) Please add the following statement, or similar, to the Methods: "This study is reported as per the Strengthening the Reporting of Observational Studies in Epidemiology (STROBE) guideline (S1 Checklist)." When you re-submit please indicate where in the manuscript this sentence has been placed.

3) In addition please remove spaces between in text reference callouts (e.g. line 99 "...conception [2, 3]." should read, "...conception [2,3]."

4) Please define all abbreviations in figure 1 in an appropriate figure caption

PRESS

Best wishes, 

Pippa

Philippa Dodd, MBBS MRCP PhD 

PLOS Medicine